# A Magnet-Based Timing Tystem to Detect Gate Crossings in Alpine Ski Racing

**DOI:** 10.3390/s19040940

**Published:** 2019-02-22

**Authors:** Benedikt Fasel, Jörg Spörri, Josef Kröll, Erich Müller, Kamiar Aminian

**Affiliations:** 1Archinisis GmbH, 1700 Fribourg, Switzerland; 2Laboratory of Movement Analysis and Measurement, Ecole Polytechnique Fédérale de Lausanne, 1015 Lausanne, Switzerland; kamiar.aminian@epfl.ch; 3Department of Orthopedics, Balgrist University Hospital, University of Zurich, 8008 Zurich, Switzerland; joerg.spoerri@balgrist.ch; 4Department of Sport Science and Kinesiology, University of Salzburg, 5400 Hallein-Rif, Austria; josef.kroell@sbg.ac.at (J.K.); erich.mueller@sbg.ac.at (E.M.)

**Keywords:** gate crossing, alpine skiing, slalom, giant slalom, validation, performance, timing, magnet

## Abstract

In alpine skiing, intermediate times are usually measured with photocells. However, for practical reasons, the number of intermediate cells is limited to three–four, making a detailed timing analysis difficult. In this paper, we propose and validate a magnet-based timing system allowing for the measurement of intermediate times at each gate. Specially designed magnets were placed at each gate and the athletes wore small magnetometers on their lower back to measure the instantaneous magnetic field. The athlete’s gate crossings caused peaks in the measured signal which could then be related to the precise instants of gate crossings. The system was validated against photocells placed at four gates of a slalom skiing course. Eight athletes skied the course twice and one run per athlete was included in the validation study. The 95% error intervals for gate-to-gate timing and section times were below 0.025 s. Each athlete’s gate-to-gate times were compared to the group’s average gate-to-gate times, revealing small performance differences that would otherwise be difficult to measure with a traditional photocell-based system. The system could be used to identify the effect of tactical choices and athlete specific skiing skills on performance and could allow a more efficient and athlete-specific performance analysis and feedback.

## 1. Introduction

In alpine ski racing, performance is defined as the total time elapsed between the start and finish [1]. For increasing spectator attractiveness (competition setting) and to provide performance-related feedback (training setting), intermediate times are also commonly used. Traditionally, total race time and intermediate times are measured with photocells [1].

Despite being technically possible, for practical reasons, the number of photocells in a competition along a race course is usually limited to three–four. Furthermore, in slalom and giant slalom world cup races, intermediate times are usually hand triggered due to too much team staff traffic on the slope and therefore potentially containing human mistakes (personal communication with FIS (Fédération Internationale de Ski) race directors). In a training setting, wireless photocells cannot be placed arbitrarily far away since radio connection between each photocell is required for the transmission of the intermediate times to the start or finish. However, especially for performance-related feedbacks during competition and/or regular training sessions, it would be desirable to have a higher amount of intermediate times and no limitations in terms of spacing between photocells.

Race time differences between athletes have been shown to be in the order of 1%–3% of total race time [2,3], with individual section time (i.e., time between two intermediate times spaced at least one gate apart) differences reaching up to 10% [3,4]. It is generally believed that these differences arise from the different technical abilities of the skiers and their tactical choices while skiing down the race course [3]. While section times may be considered inappropriate for understanding the underlying mechanism of an athlete’s overall performance [5], they might still be important to rapidly and intuitively identify strengths and weaknesses of an athlete throughout different course sections. While the different performance parameters (e.g., instantaneous performance and energy loss) proposed, and largely discussed in previous studies [2,5,6,7,8], might explain the overall performance outcome from a scientific point of view, these concepts are difficult to translate into a “coaching language”. From a practical point of view, for example, it might therefore be more appropriate to point out to an athlete that his time for a particular section is 10% worse compared to his peers.

The ability to routinely measure multiple section times or even gate-to-gate times might therefore open up new perspectives for improving performance feedback within a training session and consequently the training quality of an athlete. Gate-to-gate times would also allow for the better quantifying of the effect of different tactical choices, e.g., the strategy of minimizing gate-to-gate times at every single gate versus minimizing the gate-to-gate times at specific gates at the cost of longer gate-to-gate times at the other gates.

Especially for the technical disciplines such as slalom and giant slalom, the ski racers are passing the gates closely (approximately at distances of less than 1 m). Thus, recording the time of each gate crossing could replace the intermediate times and allow a more detailed performance feedback to the ski racers. Such a system was first described by Lachapelle et al. in 2010 [9], however, without providing any details on how timing information was obtained or validated. In 2011, Supej and Holmberg [10] published a validation study for a system to measure gate-to-gate times. In that work, skiing trajectory was measured with a differential global navigation satellite system (GNSS) fixed to the ski racers and by surveying all gate positions. Gate crossing time was then defined as the instant of time the athlete crosses a plane perpendicular to the skiing line and aligned with the gate’s position. They reported an intermediate time difference to a photocell-based system of 0 ± 3 ms (mean ± standard deviation). A similar GNSS-based approach was also suggested for sprint running [11]. Intermediate time errors over sections of 20 m of 2 ± 5 ms (mean ± standard deviation) were reported. However, in these studies, athletes were required to wear additional equipment in a small backpack or belt pocket, a GNSS reference base station was needed, and for the skiing study, an exact surveying of the gate positions had to be performed. Thus, such methods are limited regarding their practicability for use during competitions and/or daily trainings.

In contrast, magnetometers might offer a simpler approach for measuring gate crossing and intermediate times. From physical laws it is known that the magnetic field potential of a magnetic dipole is decreasing with the third power of the distance to the source [12]. If the dipole’s intensity is known, a magnetometer could be used to estimate its distance to the dipole’s centre based on the measured magnetic field. If the magnetometer is moving in space, it should be able to detect the presence of a magnetic dipole, given that it moves temporarily within a range where the magnetic field is sufficiently large to be measurable. Further, the measured magnetic field would be largest when the magnetometer is closest to the magnetic dipole. Thus, the assumptions for this study were: (1) That a magnetometer fixed to an alpine ski racer could detect the presence of magnets placed along the slope and (2) that the measured magnetic field would be at the maximum when the ski racer comes closest to the magnet. If the magnets would be placed at the gates, then the time of the gate passage could be measured. A proof-of-concept of such a system has been presented at a conference [13], however, without providing the algorithm implementation details and with a sensor fixed on the thigh, causing a turn-dependent detection bias.

Therefore, the main aim of this study was to improve the design of Reference [13] and to validate this simple-to-use magnet-based timing system to detect gate crossings in alpine ski racing. The secondary aim was to illustrate the potential for performance analysis that the presented system could offer by visual analysis of each athlete’s gate-to-gate time performance differences with respect to the group’s average gate-to-gate times.

## 2. Materials and Methods

### 2.1. Setup and Protocol

Eight male junior alpine ski racers (19.3 ± 2.5 years, 177.9 ± 4.6 cm, 74.9 ± 5.7 kg, 53.47 ± 16.72 slalom FIS-Points) participated in the study. Each of them skied twice at a predefined regular 25-gate slalom course (gate distance: 10 m; gate offset: 3 m). An inertial sensor (Physilog IV, Gait Up SA, Switzerland) was fixed on the ski racer’s lower back, at the L4–L5 level. The sensor also contained a magnetometer (MLX90393, Melexis, Belgium) sampling at 166.7 Hz. The sensitivity, offset, and axis misalignment of the magnetometer was calibrated according to Reference [14]. Bar magnets were built from five smaller disc magnets of diameter 20 mm and height 10 mm (S-20-10-N, Supermagnete, Uster, Switzerland) and separated by steel bars of diameter 16 mm and height 40 mm (Figure 1). To hold the smaller magnets and steel bars in place they were inserted into a plastic tube. The bar magnets were then placed at each gate of the slalom course (Figure 2). The magnets were inserted vertically into the snow such that the top was slightly below the snow surface in order to minimize the risk of injury. The magnets were placed such that their magnetic South poles were pointing upwards. Preliminary laboratory measurements showed that such a bar magnet was significantly distorting the ambient (Earth) magnetic field up to a distance of about 1 m with respect to the position of its magnetic South pole. This study was approved by the ethics committee of the Department of Sport Science and Kinesiology at the University of Salzburg (EC_NR. 2010_03).

### 2.2. Gate Crossing Detection Algorithm

#### 2.2.1. Signal Conditioning

For the detection of the ski racers crossing the gate, inertial (i.e., acceleration and angular velocity) and magnetometer data were used. The hypothesis was that the created distortions of the buried magnets can be detected with the magnetometer and that at the instant of gate crossing the recorded magnetic fields’ intensity becomes maximal. Since the ski racers were skiing at relatively fast speeds of over 14 m/s, a typical distortion was detectable during 0.14 s, resembling a sharp peak with a height that depends on the distance between the sensor and magnet. Thus, to detect the ski racer’s gate crossings, theoretically, it would be sufficient to detect all peaks in the norm of the measured magnetic field. However, despite careful sensor calibration, the measured magnetic field intensity (i.e., norm) was observed to be dependent on the sensor orientation (soft-iron errors) (Figure 3, top). Accordingly, prior to detecting any peaks, the measured magnetic field had to be pre-processed. As a first step, the data was up-sampled from 166.7 Hz to 500 Hz using linear interpolation. Second, the sensor’s orientation in an Earth-fixed global frame was computed with strapdown integration and static drift correction at the start and end of the run, as described in Reference [15]. The global frame’s axes were defined to be equal with the sensor axes at the beginning of the integration (this global frame was allowed to be different for each run). Third, based on the estimated orientation, the measured magnetic field was converted to the global frame. Fourth, to reduce the orientation-dependent soft-iron error and influence of any remaining orientation drift after strapdown, the magnetic field along each sensing axis was high-pass filtered using a 2nd order Butterworth filter (cut-off frequency: 1.5 Hz, corresponding to the highest expected turn frequency). Finally, the norm of this high-pass filtered magnetic field was computed. This new signal was denoted as $B_{high}\left(t\right)$.

#### 2.2.2. Peak Detection

As the high-pass filter did not only reduce soft-iron effects, but also increased the noise and decreased the relative peak heights (Figure 3, centre), the sensitivity and specificity of the peak detection performed was maximized as follows: In order to avoid missing any gate crossings when the ski racer is approaching the limits of measurable distortion, a valid gate crossing was defined to cause a peak in $B_{high}\left(t\right)$ that is higher than the 85th percentile of $B_{high}\left(t\right)$ (computed over the entire run) for at least 0.025 s. Moreover, to determine the precise instant of the maximal distortion, $B_{high}\left(t\right)$ was low-pass filtered by convolving it with a triangular window of length 0.04 s. The instant of gate crossing was then defined to happen where this low-pass filtered $B_{high}\left(t\right)$ was maximal (black circles in Figure 3, bottom).

### 2.3. Reference System and Error Analysis

The proposed system was validated against a standard time keeping system based on photocells (Witty System, Microgate, Italy). Four photocells were installed 10 cm above gates 11–14 (Figure 2). This setup allowed to obtain gate-to-gate times for gates 11–14 with a resolution of 1 ms. The section time was defined here as the time elapsed between crossing gates 11 and 14 and was also obtained with a resolution of 1 ms.

To guarantee error independence only one run for each athlete was used for validation, resulting in a dataset of N = 8 runs. Timing errors were defined as the proposed system’s values minus the reference system’s values and were reported as mean and the 95% error-range. The 95% error-range was defined as the range between the 2.5th and 97.5th percentiles.

### 2.4. Gate-to-gate Performance Analysis

For each gate, the average gate-to-gate time (e.g., performance) of all athletes and runs was computed. Then, the gate-to-gate performance difference was computed by subtracting this average gate-to-gate time from each athlete and the individual gate-to-gate run times. Finally, these performance differences were summed to obtain the total cumulated performance difference at each gate (i.e., the intermediate time with respect to the average run time at each gate was obtained). Time zero was defined as the moment the athlete crossed the first gate or the second gate if the first gate crossing was not detected.

## 3. Results

In total, 16 runs were recorded. Out of the 400 gate crossings, the proposed system detected 389 gates. Gate 1 was not detected in five runs, gate 25 was not detected in six runs, while all other gates were always detected. Gate-to-gate time for all 16 runs was, on average, 0.899 s (0.062 s standard deviation) and showed a tendency to decrease towards the end (Figure 4). For the eight runs considered for the validation, average section time computed with the photocells was 2.664 s (0.095 s standard deviation) and was 2.656 s (0.099 s standard deviation) for the proposed system. The overall gate-to-gate time’s 95% error range was [−0.013 s; 0.014 s]. The 95% error range for section time was [−0.019 s; 0.003 s] (Table 1). 

Figure 5 shows the performance difference for two runs of two selected athletes (A4 in blue and A6 in orange) with respect to the average performance. Performance differences can be observed primarily for the last third of the run whereas performance was similar for the remaining gates.

## 4. Discussion

In this study, a system was designed and validated to automatically detect gate crossings in alpine ski racing. Magnets were placed at each gate and a magnetometer fixed to the ski racers’ lower back recorded the distortions created by these magnets. We proposed a peak detection algorithm besides an inertial sensor based magnetic field correction to reach high enough precision and accuracy to show the usefulness of the method for performance evaluation. 

We reached 95% error intervals for gate-to-gate and section times below 0.025 s with respect to the reference system (Table 1). This is higher than the current alternative where a 95% error interval below 0.01 s was reached based on the skiing trajectory and a complex differential GNSS setup [10]. The higher error of our system might be explained by the relatively low sampling rate of only 166.7 Hz, limiting the time resolution of our system to 0.006 s (corresponding to a distance travelled of 0.084 m when skiing at 14 m/s), whereas Reference [10] used a spline interpolation to obtain a position measurement at every 0.001 m. Thus, by increasing the magnetometer’s sampling frequency, the system’s errors might be further reduced. 

However, compared to the GNSS-based approach, the proposed system’s setup is easier to use since it does not need a differential GNSS with a base-station and a surveying of each gate’s position. Except for the magnets, no other extra hardware is required to be placed on the ski slope. Meaningful performance-related time differences within short sections have been reported to be in the range of 0.02 s and more [2,3,4,5,16,17]. Therefore, our system might be limited in reliably detecting time differences for very subtle performance changes. On the other hand, the proposed system is easy to use and proved to have an even lower setup time than photocells. Therefore, although maybe slightly less precise than photocells, such a system could find a broad acceptance among coaches for a regular use during daily trainings. To increase practical use in a training environment, collected data could be automatically downloaded over Bluetooth after each run and processed within a smartphone app. For real-time applications in a race environment, the collected data could be processed on the sensor and results wirelessly transmitted to base stations installed along the ski slope.

Another advantage of the proposed system was that timing information could be obtained for every single gate of the course, allowing a much more in-depth performance analysis for each athlete compared to a photocell-based time measurement system. Figure 5 illustrates well the information gain of the gate-to-gate timing. Both athletes were better than the average, however, in different ways. Ignoring a performance difference at the start, the athletes had a similar performance increase during the first two thirds of the race. However, for the last third, A6 gained in both of his runs significantly more time, while A4 was losing time. A traditional timing system with only one or two intermediate times over the 25 gates would not have been able to measure this difference in performance. For the practitioner, such an analysis might thus provide essential information about how the technical skills and tactical choices of each athlete influence the performance.

### 4.1. Potential System Limitations

At first glance, a potential limitation of the proposed magnet-based timing system might be the fact that the detectable distance between the magnetometer and gate is given by the magnet’s specifications. The current setup allowed a maximum distance of approximately 1 m. Thus, it can only be used for the detection of gates at which the ski racers are passing closely. As a consequence, in the current study, not all gate crossings for the first and last gates could be detected. While in slalom and giant slalom skiing the athletes usually cross each gate closely, in super-G or downhill skiing the current setup might not work in all race situations. However, in such cases, multiple magnets could be buried under the snow surface along a perpendicular line that crosses the skiing line. Such a setup could also be used for marking the start and finish lines. Another potential limitation of the proposed timing system might be related to the fact that time during which a distortion is measured decreases for increased speed. Consequently, the peak-width is also decreased. However, the sampling frequency and peak detection algorithms were designed to work for speeds of up to 140 km/h allowing for measurement of the gate crossings for all skiing disciplines. Similar to what is already reported in Reference [10], it is expected that timing precision will increase for higher speeds, mostly due to the reduced peak-width, and thus, decreased uncertainty in the estimated peak location.

### 4.2. Limitation of the Study

A limitation of the present study was the small sample size of only eight athletes. However, 100% of the gates (excluding the first and last gates) were detected correctly and no outliers were observed, confirming the validity of the results. Nevertheless, for future applications in competitions, we advise the use of a traditional photocell-based system in parallel to allow a critical comparison with the proposed system. 

Finally, it has to be pointed out that a major advantage compared to traditional photocell-based timing systems is that the proposed magnet-based system is not limited by the number of intermediate times and cannot have any wrong triggering caused by other team staff on and around the track. Thus, with the proposed system, gate-to-gate time performance analysis can be implemented and ski racers’ performance can be tracked and compared at every gate along the entire course.

## 5. Conclusions

The proposed magnet-based timing system allowed for the accurate and automatic detection of ski racers’ gate-to-gate times during a slalom race. More gates could be measured with a minimum of setup efforts compared to a traditional photocell system. To further reduce setup efforts, the magnet could be directly integrated into the gates’ base. The errors were in the range of the minimum time resolution typically required for performance analysis in skiing (0.02 s), and therefore, the system could also be used instead of photocells during regular training sessions. The proposed setup (permanent magnet’s strength and placement) may need to be adapted for super-G and downhill skiing where athletes may not cross the gates at a sufficiently small distance to record the magnetic field disturbance. The system allows for the identification of the effect of tactical choices and athlete specific skiing skills on performance and may therefore help the coaches to do a more specific performance analysis and feedback.

## Figures and Tables

**Figure 1 sensors-19-00940-f001:**
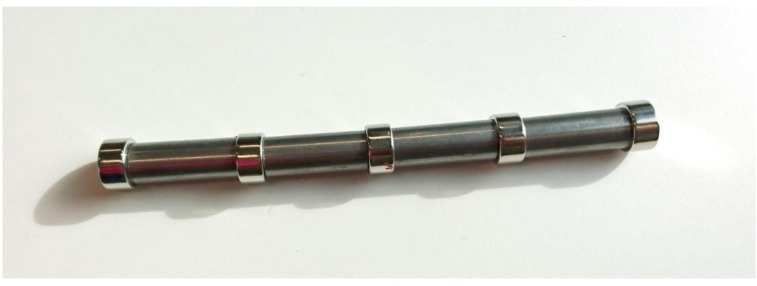
Photograph of the bar magnet that was built from five smaller disc magnets and separated by steel bars.

**Figure 2 sensors-19-00940-f002:**
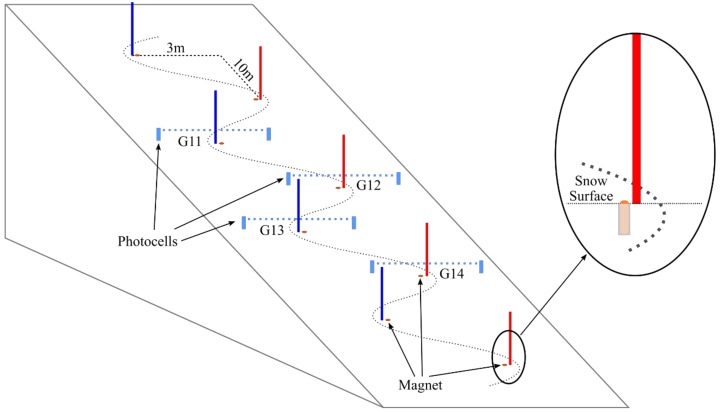
Setup of the slalom course. For better visualization, only gates 9 to 16 are shown. The photocells of the reference system were placed 0.1 m before gates 11–14. The bar magnets were buried completely into the snow surface to avoid any risk of injury.

**Figure 3 sensors-19-00940-f003:**
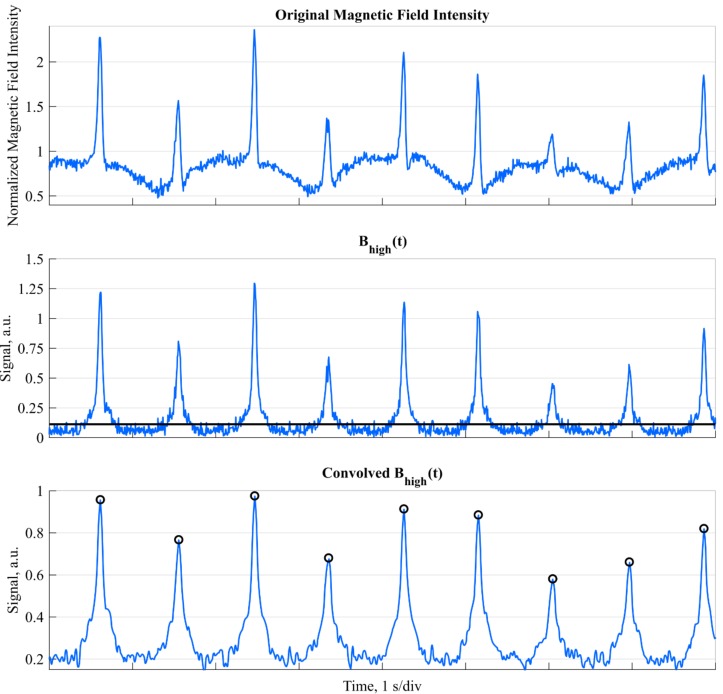
Original magnetic field intensity (top), high-pas filtered field intensity $B_{high}\left(t\right)$ (centre), and convolved $B_{high}\left(t\right)$ (bottom). The figure shows the gate crossings of 9 consecutive gates. The soft-iron errors were visible on the original magnetic field intensity (oscillating “baseline” signal). In $B_{high}\left(t\right)$ these “baseline oscillations” were removed; however, at the cost of reduced peak height and increased signal noise. The black line shows the 85th percentile. In the convolved $B_{high}\left(t\right)$, the relative peak height was increased and the signal noise was reduced, allowing a more precise detection of the gate crossing events (marked with the black circles).

**Figure 4 sensors-19-00940-f004:**
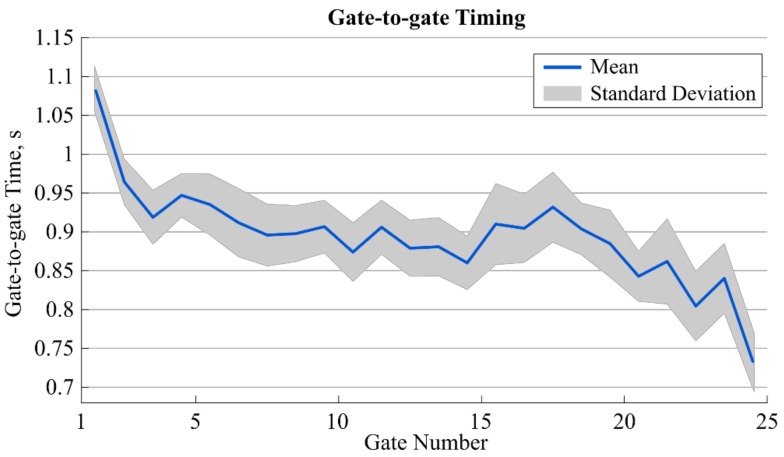
Mean (solid line) plus/minus one standard deviation (shaded area) for the gate-to-gate timing for all gates and the 16 runs. The value at position i+0.5 refers to the gate-to-gate time between gates i and i+1.

**Figure 5 sensors-19-00940-f005:**
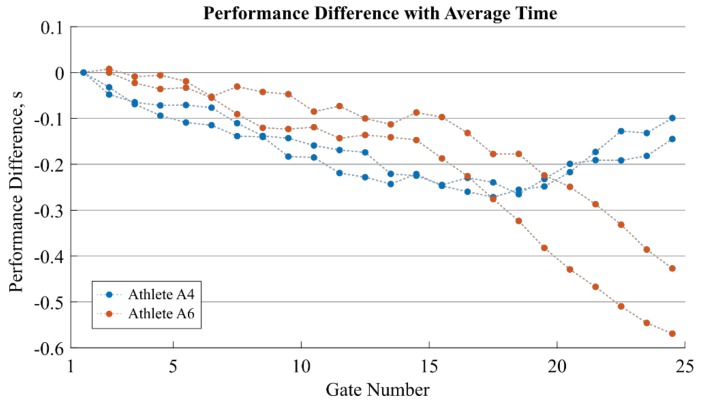
Performance gain/loss analysis compared to average group performance. The cumulated performance difference to average group performance is shown for the two runs of athlete A4 (in blue) and A6 (in orange). Both athletes were faster than the group average. Each dot corresponds to one gate-to-gate time measurement where the value at position i+0.5 refers to the gate-to-gate performance difference between gates i and i+1. A negative performance means that the athlete was faster than the average performance. A negative slope means a gain of performance, whereas a positive slope means a loss of performance.

**Table 1 sensors-19-00940-t001:** Errors of the magnet-based system with respect to the photocell system.

	Errors, s			
Athlete	Gate 11 to 12	Gate 12 to 13	Gate 13 to 14	Section time
A1	0.001	−0.002	−0.007	−0.008
A2	−0.009	−0.005	−0.004	−0.018
A3	0.008	−0.008	−0.002	−0.002
A4	0.005	−0.010	−0.006	−0.011
A5	0.001	−0.013	−0.007	−0.019
A6	0.014	0.006	−0.005	0.003
A7	0.003	−0.010	−0.002	−0.009
A8	−0.007	0.000	0.010	0.003
Mean error	0.002	−0.007	−0.003	−0.008
2.5th percentile	−0.009	−0.013	−0.007	−0.019
97.5th percentile	0.014	0.000	0.010	0.003
95% error interval	0.025	0.013	0.017	0.021

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
