# Peer review of "A Magnet-Based Timing System to Detect Gate Crossings in Alpine Ski Racing"

_sensors, 2019, doi:10.3390/s19040940_

Reviewer 1 Report

The paper is fine as is with a minor comment: The Author should consult the following paper and refer to it in their paper as it was the first paper [to my knowledge] that describes the development and testing of a light weight GNSS system usable by skiers and capable of delivering continuous cm-level accuracy [and timing to better than 1ms]  without markers along the slope:
Lachapelle, G., A. Morrison and R. Ong (2010) STEALTH™ - A GNSS-Based Advanced Device to Train Canadian Olympic Skiers. Geomatica, 64, 3, 327-335. A version of above should be available on PLAN.geomatics.ucalgary.ca

Author Response

Dear reviewer,
Thank you for your review and pointing out this publication. We added it in the introduction. The 5th paragraph (starting at line 61) was changed and now reads: ... Such a system was first described by Lachapelle et al. in 2010 [9], however, without providing any details on how timing information was obtained or validated. In 2011, Supej & Holmberg [10] published a validation study for a system to measure gate-to-gate times. ...

Reviewer 2 Report

The authors describes the mechanism of the magnetic sensing and the experimental environments and the methods in a fluent presentation. However, it is not clear to know how to measure the sensing data shown in figure 3. It would be better to add a short explanation of a detection box (i.e. a small system with memory to store the sensing data) with a photo attached to ski. If the explanation of the box was described, it is very clear how the data shown in figure 3 is measured and the characteristics because the information of attached place in ski induce the intuitive imagination of the result in figure3.

Author Response

Dear reviewer,
Thank you very much for your review and feedback. The sensor we used is described in section 2.1 Setup and Protocol:
"An inertial sensor (Physilog IV, Gait Up SA, Switzerland) was fixed on the ski racer’s lower back, at the L4-L5 level. The sensor also contained a magnetometer (MLX90393, Melexis, Belgium) sampling at 166.7 Hz".
Since this is a commercial device where product pictures and datasheets are available on internet (https://gaitup.com/wp-content/uploads/Brochure_Datasheet_Physilog_RA_V2.6.pdf) we decided not to add any more descriptions or pictures to the manuscript.

Reviewer 3 Report

In this paper, the authors have validated a novel magnet-based timing system to determinate section time and gate-to-gate times in alpine ski racing. The proposed setup was composed by inertial sensors placed on subjects’ lumbar region and bar magnets closely placed to each gate of the slalom skiing course. A peak detection based algorithm was developed to determinate gate crossing. Timing error was assessed considering a photocell systems as a reference. The manuscript is well written and conveys a clear message.  However, it is not completely clear how the authors envision their method being used in real-time athlete performance monitoring. Apparently, the peak-detection method identifies gate-to-gate crossings at the end of each run, while photocell based timing system provides real-time information on gate-to gate and/or section speed. More details explanation on how authors intentions foresee this semi-real-time monitoring of performance training should be reported. Moreover, authors should report a briefly description on their previous work focused on similar scope, highlighting differences in sensor positioning and in results.
· Fasel, B., Spörri, J., Kröll, J., & Aminian, K. (2016). Alpine ski racing gate crossing detection using magnetometers. In Proceedings of the 7th International Congress on Science and Skiing, Salzburg.
Other issues:
1. Page 1 line 31-35: Please add references to these sentences.
2.Page 2 line 46: Authors mentioned the terminology “section time” for the first time in the introduction section. Please add the definition in this section.
3. Page 4 line 135: Please justify the choice of cut-off frequency. Did authors perform a previous analysis on the most suitable filtering process?
4.Page 5 line 177: Please replace “sec” with “s” as the International System Unit recommendations require. Please replace it on the entire manuscript.

Author Response

Dear reviewer,
Thank you very much for your review and comments.
We adapted the manuscript as follows:
- Real-time application. The following sentence was added to the discussion: "To increase practical use in a training environment, collected data could be automatically downloaded over Bluetooth after each run and processed within a smartphone app. For real-time applications in a race environment, the collected data could be processed on the sensor and results wirelessly transmitted to base stations installed along the ski slope."
- Contrast to previous work: The following sentence has been added to the introduction: "A proof-of-concept of such a system has been presented at a conference [13], however without providing the algorithm implementation details and with a sensor fixed on the thigh, causing a turn-dependent detection bias."
- Lines 31-35 misses references: We could not find any scientific paper on a study on which timing systems are used how much in a skiing training setting. From our own experience with different national teams, the use of photocells is very standard (used in almost every training and on all skill levels) and no other timing systems are used (yet). We therefore think that using photocells for measuring time in alpine ski racing is "common knowledge" and does not need further references.
- Line 46: We added the following text at the first mentioning of section time, at line 44: "(i.e. time between two intermediate times spaced at least one gate apart)"
- Line 135: We chose this cut-off frequency as being slightly higher than the movement frequency due to the turns (usually turns are spaced more than 0.75 sec apart). We added the words "corresponding to the highest expected turn frequency" in parantheses after the mentioning of the 1.5 Hz cut-off frequency.
- Line 177: Replacing sec with s: We replaced the units in the entire manuscript and all figures and tables.